# Human Milk Concentrations of Minerals, Essential and Toxic Trace Elements and Association with Selective Medical, Social, Demographic and Environmental Factors

**DOI:** 10.3390/nu13061885

**Published:** 2021-05-31

**Authors:** Natalia Mandiá, Pilar Bermejo-Barrera, Paloma Herbello, Olalla López-Suárez, Jose M. Fraga, Cristina Fernández-Pérez, María L. Couce

**Affiliations:** 1Department of Neonatology, University Clinical Hospital of Santiago de Compostela, 15704 Santiago de Compostela, Spain; olalla.elena.lopez.suarez@sergas.es; 2IDIS-Health Research Institute of Santiago de Compostela, 15704 Santiago de Compostela, Spain; josemaria.fraga@usc.es; 3Faculty of Medicine, University of Santiago de Compostela, 15704 Santiago de Compostela, Spain; 4Group of Trace Elements, Speciation and Spectroscopy (GETEE), Strategic Grouping in Materials (AEMAT), Department of Analytical Chemistry, Nutrition and Bromatology, Faculty of Chemistry, University of Santiago de Compostela, Avenida das Ciencias, s/n, 15782 Santiago de Compostela, Spain; pilar.bermejo@usc.es (P.B.-B.); paloma.herbello@usc.es (P.H.); 5Department of Preventive Medicine, University Hospital of Santiago de Compostela, Santiago de Compostela University, 15704 Santiago de Compostela, Spain; cristina.fernandez.perez3@sergas.es; 6MetabERN, via Pozzuolo 330, 33100 Udine, Italy

**Keywords:** breast milk, trace elements, minerals, toxic metals, infant milk formula, newborn, preterm

## Abstract

This study aims to quantify concentrations of minerals and trace elements in human milk (HM) and infant formula (IF) and evaluate associations with medical, social, environmental, and demographic variables. A prospective, case series study of 170 nursing mothers was made. HM samples were obtained from full-term (colostrum, intermediate and mature HM) and preterm (mature HM) mothers. Variables of interest were assessed by a questionnaire. For comparison, IF samples (*n* = 30) were analyzed in a cross-sectional study. Concentrations of 35 minerals, essential and toxic trace elements were quantified, 5 for the first time: thallium in HM and IF; strontium in preterm HM; and gallium, lithium and uranium in IF. In preterm and full-term HM, levels of selenium (*p* < 0.001) were significantly lower than recommended and were associated with low birth weight (*p* < 0.002). Cesium and strontium concentrations were significantly higher than recommended (*p* < 0.001). Associations were observed between arsenic and residence in an urban area (*p* = 0.013), and between lead and smoking (*p* = 0.024) and well-water consumption (*p* = 0.046). In IF, aluminum, vanadium, and uranium levels were higher than in HM (*p* < 0.001); uranium, quantified for the first time, was 100 times higher in all types of IF than in HM. Our results indicate that concentrations of most trace elements were within internationally accepted ranges for HM and IF. However, preterm infants are at increased risk of nutritional deficiencies and toxicity. IF manufacturers should reduce the content of toxic trace elements.

## 1. Introduction

Human milk (HM) is considered the gold standard for infant nutrition, both for full-term and preterm infants [1,2]. HM influences the intestinal microflora, ensures structural and functional maturity of the mucous membranes, reduces the risk of allergies and autoimmune disorders, and contributes to proper development of the digestive, central nervous, endocrine, and immune systems [3,4,5]. The composition of HM is not always the same, the fat and energy content varies from the beginning to the end of the HM intake, it follows a diurnal pattern and varies between each individual, depending on the type of delivery, lactation period, maternal diet and area of residence [6,7]. It is widely reported that maternal diet influences the nutritional composition of breast milk [8]. However, the amount of variability in HM attributable to diet remains mostly unknown. Previous studies on trace elements in HM included factors affecting its trace elements and maternal diet. Most studies have focused on component analysis or nutritional aspects of HM, but only a few studies have confirmed the relationship between trace elements in HM and psychosocial variables [9]. Donated human milk (DHM) can be used in preterm infants when HM is insufficient or not available. 

For babies that cannot be breastfed or receive DHM, one alternative is infant milk formula (IF), the composition of which is continuously adapted to provide similar nutritive benefits to HM. Recommendations on the composition of IF and HM are established by the European Society for Paediatric Gastroenterology Hepatology and Nutrition (ESPGHAN) [10] and the American Academy of Pediatrics (AAP) [11]. 

Deficits in micronutrients in HM or IF in early life have adverse effects on infants and are associated with short-term infections and higher rates of diseases [12]. Excessive levels of micronutrients can also be harmful [6,13]. In addition to essential elements, milk consumption can also result in the transfer to infants of potentially toxic metals [14]. Given the importance of adequate micronutrient intake in early life and the differences in diets and environments between populations, analysis of trace elements in IF is important from a public health perspective. In recent years, several studies have measured the concentration of trace elements in HM from women in different countries [1,15,16,17,18,19,20]. However, no study has comprehensively compared the levels of all essential elements in HM versus IF. To evaluate the composition of milk in our population and potential health risks associated, we quantified levels of minerals and trace elements in HM samples acquired at different stages from mothers of preterm and full-term infants and in samples of IF for infants in the first year of life.

## 2. Material and Methods 

### 2.1. Study Design and Population

We conducted a prospective, case series, single-center study of nursing mothers and a cross-sectional study of IF in University Clinical Hospital of Santiago de Compostela (Spain), shown in Figure 1. The inclusion criteria of nursing mothers were maternal age > 18 years old, without chronic disease and without taking nutrient supplements. All potential participants were introduced to this research and invited to join the study during prenatal and postnatal care at our institution. After receiving prior written informed consent, HM samples (5–10 mL) were obtained in 3 different periods of lactation during the first 6 months after birth: colostrum during the first 3–4 days of lactation, intermediate milk up to 7–10 days, and later mature milk, both in mothers of full-term; and later mature in mothers of premature newborns. Samples of full-term colostrum (*n* = 70), intermediate HM (*n* = 70), mature HM (*n* = 70) and preterm mature HM (*n* = 100) were collected between 1 January 2018 and 30 June 2019. In addition, we made a comparison group with 30 IF samples, selecting the brands used in our institution for children under 1 year of age, that account for 30% of the total brands sold in Spain, and classified into 4 groups: starter formulas (*n* = 13), continuation formulas (*n* = 10) (both milk protein-based formulas), hydrolyzed formulas (*n* = 5), and formulas for preterm infants (*n* = 2). 

Concentrations of the elements in milk were analyzed at the Laboratory of Analytical Chemistry, Nutrition and Bromatology of the University of Santiago de Compostela. Elements were classified into 3 groups: minerals (*n* = 5), including calcium (Ca), potassium (K), magnesium (Mg), sodium (Na), phosphorus (P); essential trace elements (*n* = 9), including cobalt (Co), chromium (Cr), copper (Cu), iron (Fe), iodine (I), manganese (Mn), molybdenum (Mo), selenium (Se), zinc (Zn); and toxic trace elements (*n* = 21), including silver (Ag), aluminum (Al), arsenic (As), barium (Ba), beryllium (Be), cadmium (Cd), cesium (Cs), gallium (Ga), mercury (Hg), lithium (Li), nickel (Ni), lead (Pb), platinum (Pt), rubidium (Rb), antimony (Sb), tin (Sn), strontium (Sr), titanium (Ti), thallium (Tl), uranium (U), and vanadium (V). 

Data were collected on medical, social, environmental, and demographic factors that may influence the composition of HM. For each participating mother, age, weight gain during pregnancy (excessive weight gain ≥16 kg) [21], residency, and smoking and drinking status were evaluated at the end of pregnancy. Gestational age and birth weight were recorded for all newborns. 

Study approval was obtained from the Research Ethics Committees of Galicia (2017/082) and all the samples collected were analyzed exclusively for the purpose of the present study.

### 2.2. Method

#### 2.2.1. Sample Collection and Preparation

For collection of HM samples, the nipple area of the breast was washed with soap and water and HM was manually extracted and collected in sterile plastic containers made of polyethylene terephthalate (PET). The containers were labeled to indicate the day of sample collection and all samples were stored at −20°C until analysis. For IF samples, 30 g of milk powder was collected in PET storage containers under a laminar flow hood and subsequently reconstituted following the manufacturer’s recommendations.

#### 2.2.2. Analyses

Levels of trace elements in milk samples were quantified using inductively coupled plasma mass spectrometry (ICP-MS), following the procedure described by Mohd-Taufek et al. [22]. For this, a solution is prepared containing 0.01% (m/V) of Triton X-100, 10 g/L of Ethylenediaminetetraacetic Acid (EDTA), 2.5% (*v*/*v*) of ammonia and 10% (*v*/*v*) of 2-propanol prepared in Mili-Q^®^ ultrapure water. Once the HM samples are homogenized by heating them in an ultrasonic bath between 35 and 38 °C, 400 µL of milk is taken, 1 mL of the previously prepared alkaline solution is added and it is brought to a final volume of 10 mL with Military H_2_O Q^®^.

The preparation of IF samples has been performed by simplifying the process, since the fat content of IF is lower than in HM and is also hydrolyzed, resulting in a much simpler matrix. The quantity of sample necessary to obtain the same proportion recommended by the manufacturer of the IF was weighed. Once dissolved and homogenized, samples of 0.4 mL of milk and 1 mL of the solution of 0.01% (*v*/*v*) of Triton X-100 were taken, and H_2_O Mili-Q^®^ was added to a final volume of 10 mL. The NIST SRM 1849 for IF samples and the certified milk reference materials ERM-BD 150 for low concentration levels and the ERM-BD 151 for higher levels in some elements have been used as certified reference materials. Once dissolved with ultrapure water, these materials have been prepared in the same manner as the samples.

The measurements of the trace elements in the milk samples have been performed with an ICP-MS model NexION^®^ 300× (PerkinElmer Inc., Shelton, CT, USA). The standard addition method has been used for the quantification of concentrations using different concentration levels between 0 and 25 µg/L. In the case of the majority elements, the measurement equipment has been used with an inductively coupled optical atomic emission spectrometry (ICP-OES) model Optima 3300 DV (PerkinElmer Inc., Norwalk, CA, USA). The calibration of the equipment has been carried out using the standard addition method with concentration standards between 0 and 5 mg/L for Ca, K, and Mg, and between 0 and 25 mg/L for Na and P. The instrumental conditions of the ICP-MS and ICP-OES are detailed in the Appendix A.

### 2.3. Statistical Analysis

A minimum sample size of 63 per group was required to detect differences of at least 50% between the means of two normal quantitative variables with a significance level of 5% and a statistical power of 80% in the case of human milk samples. For the infant formula samples, the sample size was not calculated. Data were analyzed using the statistical program SPSS. Categorical variables are presented as numbers and percentages, and continuous variables as the mean and standard deviation. Normality was assessed using the Shapiro–Wilk test. For normally distributed numeric variables, ANOVA was used to compare groups. In addition, in order to compare the means in groups of different sizes, the Bonferroni test was used. Categorical variables were compared using the χ^2^ test. Associations between absolute change means in trace element concentrations and the variables of interest were evaluated using linear regression models, with change represented as a coefficient. Results are presented with the corresponding 95% confidence interval and *p*-values < 0.05 indicate a statistically significant difference. 

## 3. Results

### 3.1. Characteristics of the Study Participants

Table 1 shows medical, social, environmental, and demographic data for participating mothers and their infants, according to type of delivery (preterm or full-term). Comparing the characteristics of the two groups, full-term mothers and preterm mothers, both are homogeneous in terms of no significant differences except in the mean gestational age (39 vs. 31 weeks, *p* < 0.05) and the birth weight of the newborns (2990 g in term deliveries vs. 1445 g in preterm deliveries, *p* < 0.05).

### 3.2. Minerals and Trace Elements in Human Milk

The concentrations of 35 elements in HM are shown in Table 2. These data include Tl levels (mean concentration, 0.04 ± 0.05 µg/L in mature HM), which have not been previously quantified in HM. Ca concentrations were higher in mature HM compared with colostrum (*p* = 0.006) and preterm HM (*p* = 0.024) compared with mature term HM. Analysis of essential trace elements revealed significantly lower levels of Cu, Mn, Mo, and Se in full-term mature HM (*p* ≤ 0.039), and of Cr, Fe, I, Se, and Zn in preterm HM (*p* ≤ 0.045). Notably, Se levels were below those recommended by international standards (Appendix A). 

Levels of the toxic elements Cs, Pt, Sn, and Sr were significantly increased (*p* < 0.050) in full-term colostrum, while those of Cd, Cs, Ga, Hg, Sb, Sr, Ti, and V were significantly increased in preterm mature milk (*p* ≤ 0.047). These increases were particularly notable for Cs and Sr, levels of which were up to two times higher than those considered acceptable by the AAP (Appendix A). For four specific elements we observed significant differences in concentrations at each of the timepoints at which preterm and full-term HM was sampled: the mineral Ca; the essential element Se; and two toxic elements, Cs and Sr (Figure 2).

### 3.3. Minerals and Trace Element in Infant Formula

The concentrations of 35 elements analyzed in IF samples are presented in Table 3. For the four different types of IF analyzed, trace elements were within the range recommended by ESPGHAN (*n* = 12) (Appendix A), and were consistent with the information provided by the respective manufacturers (Appendix A). Of 35 elements quantified, concentrations in IF have been previously reported for 31, the levels of the four remaining elements are reported here for the first time: Ga, Li, Tl and U.

Fe is the only essential trace element for which we detected an increase in continuation IF (*p* = 0.027) compared with starter IF. In preterm IF we observed a significant increase in Co (*p* < 0.001), levels of which were up to 3–6 times higher than in the other IF types analyzed. Preterm IF also contained significantly higher concentrations of the toxic elements Sr, U, and V (*p* ≤ 0.019). 

As shown in Table 4, compared with mature HM we observed increases in the concentrations of toxic trace elements Al, Be, Rb, Sr, U and V in starter and continuation IF (*p* ≤ 0.001). In addition, they are also increased in preterm IF compared to preterm HM. This effect was particularly notable for U, levels of which were over 100 times higher in all types of IF (*p* < 0.001).

### 3.4. Associations with the Medical, Social, Environmental, and Demographic Variables

All participating mothers (70 full-term and 100 preterm delivery) completed a questionnaire, the responses to which were analyzed by multivariate analysis (Table 5). The results revealed a positive correlation between excessive body weight gain during pregnancy and HM concentrations of Na, Fe, and I (*p* < 0.050), and a negative correlation between birth weight and most essential (*n* = 6) and toxic elements (*n* = 9) (*p* < 0.029). Moreover, we observed significant positive correlations between HM concentrations of Ca and Na and gestational hypertension (*p*≤ 0.024), and between residence in urban environment and higher levels of As (*p* = 0.013). This examination also revealed a positive correlation between well-water consumption and HM concentrations of Na, Cu, Fe, Pb, and Ti (*p* ≤ 0.046). Finally, there was a significant positive correlation between smoking and HM concentrations of Ba and Pb (*p* < 0.050).

## 4. Discussion

### 4.1. Minerals and Essential Trace Elements

Our findings reveal a trend towards an increase in Ca concentrations between colostrum and mature milk, in agreement with the findings of Prentice and Barclay [23]. As previously reported by Atkinson et al. [24] and Schanler [25], Ca levels were higher in preterm versus mature full-term HM. Decreases in the concentrations of essential trace elements such as Cu, Mn, Mo, and Se were observed as lactation progressed, probably due to decreases in the levels of proteins in milk that serve as ligands of trace elements [26,27]. We also observed significant decreases in Cr, Fe, I, Se, and Zn concentrations in premature HM compared with mature full-term HM. This effect was particularly notable for Se, levels of which were lower than those reported in other studies [28,29]. This observation may be partially due to the low levels of Se in soil in Spain [30], the country of origin of the mothers participating in the study. In another study, two Portuguese selenium-rich regions and a control region in Yaracuy state were compared. A significant increase in selenium was observed, from 42.9 μg/L for the control region in Yacuray to 56.6 and 112.2 μg/L for the two seleniferous regions, values in all cases much higher than those found in our study and our country [31]. Reduced Se levels in HM could potentially adversely affect the functional activity of antioxidant selenoproteins, compromising protection against placental oxidative stress and detrimentally impacting fetal growth. In fact, Se deficiency has been associated with preeclampsia [32], preterm birth [33], and small for gestational age (SGA) infants [34]. Ustundag et al. reported significantly lower Zn levels in milk from mothers of preterm versus full-term babies [35]. Premature and SGA infants have higher essential micronutrient requirements due to their rapid postnatal growth and development and the limited capacity for storage of these elements [36]. These infants are therefore at increased risk of developing nutritional deficiencies.

We observed a correlation between excessive maternal weight gain during pregnancy and HM levels of Na, Fe, I, Mo, and Se. Regarding Fe and I, similar results were reported [37,38], with a positive association between HM levels and maternal weight. The higher HM levels of Na and Ca in mothers with gestational hypertension could be explained in a similar way to the increased levels of Na and Ca in mothers with arterial hypertension [39,40], as the higher the blood serum levels of Ca and Na in the mothers, the higher the levels of these elements in the breast milk. We observed no significant association between the levels of essential trace elements and area of residence, urban or rural, in line with the findings of Domellöf et al. [41]. The consistency of trace element concentrations across populations, despite geographic and lifestyle-related variations, likely reflects the importance of these trace elements for proper development and function, and common physiological mechanisms to maintain levels adequate for the infant.

As expected, our analyses of IF showed increases in Fe content in continuation IF. The AAP Committee on Nutrition has strongly advocated Fe fortification of IF as a means of reducing the prevalence of anemia and concomitant sequelae during the first year of life [42]. Essential trace elements, in particular Fe and Zn, are found at concentrations 15–50 times higher in IF than HM, as the formulation of IF must take into account losses that occur during production and storage, with the bioavailability of essential trace elements being much lower than in breast milk [13].

### 4.2. Toxic Trace Elements

Our analyses include the first reported quantification of the levels of Tl in HM, which revealed a mean concentration of 0.04 ± 0.01 µg/L in mature HM. Tl is a highly toxic metal that is also found naturally in the environment, and therefore can contaminate water and food [43]. To date, no tolerable daily intake has been defined.

The concentrations of toxic metals such as Cs, Pt, Sn, and Sr decreased significantly in mature HM. The basis for this observation remains unclear, although HM proteins have a high capacity for binding toxic metals, being highest in colostrum.

Previous studies have reported that exposure of the mother to Cd may increase the likelihood of preterm delivery and, consequently, low birth weight [44]. Significantly higher levels of Hg have been reported in HM from mothers of preterm babies compared with full-term HM [45]. These observations are in agreement with the findings in our study population, in which Cd, Cs, Ga, Hg, Sb, Sr, and Ti concentrations were higher in HM from mothers of preterm babies compared with full-term HM; in addition, this is the first time that Sr levels were analyzed in premature HM. Notably, we found that Cs levels in preterm HM were up to two times higher than those reported in previous studies [46]. Chronic Cs ingestion has been found to cause heart failure [47]. In their analysis of bone biopsy samples, D’Haese et al. found that Sr levels were increased in patients with osteomalacia [48]. Moreover, rickets is a major problem in premature and SGA infants, suggesting a potential association with the high levels of Sr in HM consumed by these infants. However, there is little information available about the toxic effects of Sr. Cs and Sr have been distributed in the environment due to nuclear weapons testing, nuclear power production and nuclear accidents. These radionuclides are of particular concern as they are readily incorporated into biological systems due to their chemical similarity to the biologically essential elements K in the case of Cs and Ca in the case of Sr. In the long term, Cs and Sr mainly enter the human food chain by consumption of plants grown on contaminated soils or products from animals fed on contaminated fodder [49]. Although preliminary determinations had shown that contamination levels in human milk were minimal, concentrations from 1 to 5 μg/L have been described for Cs [50] and 44–46 μg/L for Sr [51].

We observed an association between higher HM concentrations of As and residence in an urban area. This could be explained by an increase in the availability of As in food, water, and air caused by industrial activities. Rahimi et al. [52] reported significant increases in Pb concentrations in HM from mothers exposed to smoking, in line with the present findings. Furthermore, studies have reported higher mean Pb levels in drinking groundwater than drinking surface water [53]. In certain areas, water pipes may still be jointed with Pb solder, and lead-lined storage tanks are common in houses [54]. These observations may at least partially explain the higher Pb concentration detected in HM from mothers that consumed well water.

Our analyses of commercial IF samples revealed high toxic trace elements, in particular Al. This finding is in line with those previously reported by our group [55]. Specifically, we reported levels of Al in hydrolyzed IF that were higher than those in other types of IF, although within the limits recommended by ESPGHAN. Al content of IF is 5–8 times higher than that of HM, a factor that contributes significantly to the body burden of Al in infants. Al has long been implicated in the etiology of Alzheimer disease (AD) [56]. Specifically, pathological concentrations of Al in the brain (>2.00 μg/g dry weight) contribute to earlier and more aggressive AD [57]. Compared with HM, we also found high levels of U and V in IF, and elevated vanadium levels are related to intestinal disorders [58].

The present study is the first to quantify levels of Ga, Li, Tl and U in IF, and revealed significantly higher U concentrations compared with HM. The main adverse health effects of U exposure are usually due to its significant chemical toxicity, which can affect neurological and reproductive systems [59]. Even though there are no significant differences with HM concentration, Ga and Tl are extensively used in advanced industries and are considered as toxic to humans. There is a growing concern about the potential release of these materials into the environment leading to effects on public and environmental health. So far, a tolerable daily (or weekly) intake has not been derived. Human exposure can take different routes: oral, by ingestion of contaminated food; dermal; or respiratory, by inhalation of dust and fumes. The most prominent feature of Tl poisoning is hair loss or alopecia [43]. Other symptoms, such as gastrointestinal disturbances, high blood pressure, rapid heart rate, and persistent weakness, are possible consequences of poisoning by these elements [60].

It is clear that the presence of toxic elements in IF accounts for a significant component of early life exposure to this contaminant, and every effort should be made by manufacturers to reduce the concentrations of these products to an achievable practical minimum.

Some limitations of the present study should be noted. The smaller sample size of the IF, compared with HM, makes it difficult to draw meaningful conclusions from the results obtained. The concentration of the trace elements of interest was not measured in maternal serum and maternal diets were not thoroughly analyzed. Ideally, follow-up of the infants fed with the HM and IF analyzed in this study should be performed to determine possible long-term consequences. The main strength of this study is the evaluation of the micronutrients composition of HM and the simultaneous assessment of the influence of mother and baby-related variables. Moreover, our analysis of trace element concentrations in IF allowed for comparison with concentrations reported by the respective manufacturers and recommended values by international standards.

## 5. Conclusions

We report for the first time the concentrations of Tl in HM and IF; and Ga, Li, and U in IF. We found Se levels in HM are below those recommended, and were associated with low birth weight, and being at risk of nutritional deficiencies. Furthermore, we observed significant increases of the concentration of the toxic trace element Cs, levels of which were double those recommended, and report for the first time the concentration of preterm HM of Sr. Our data highlight the potential influence of environmental factors on the concentrations of toxic trace elements in HM, demonstrating significant associations between As levels and residence in an urban area, and between Pb levels and both smoking and the consumption of well water. Finally, our analyses of IF indicate higher levels of Al, V, and U than found in HM. These results underscore the importance of reducing the concentrations of these toxins in IF to avoid long-term health consequences for infants.

## Figures and Tables

**Figure 1 nutrients-13-01885-f001:**
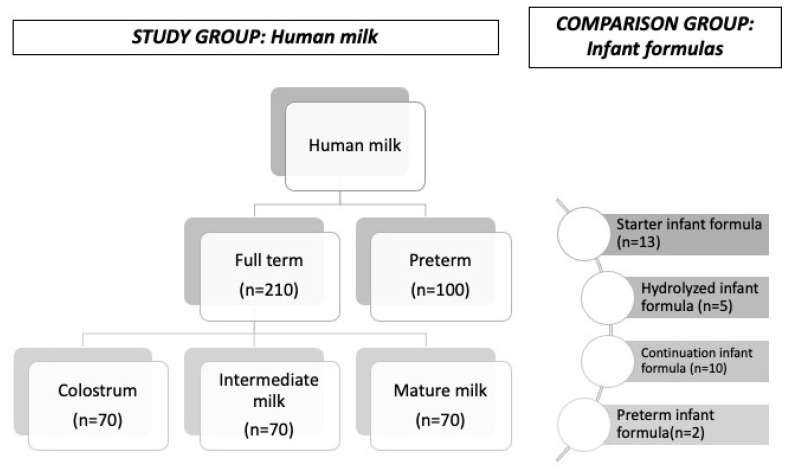
Study design.

**Figure 2 nutrients-13-01885-f002:**
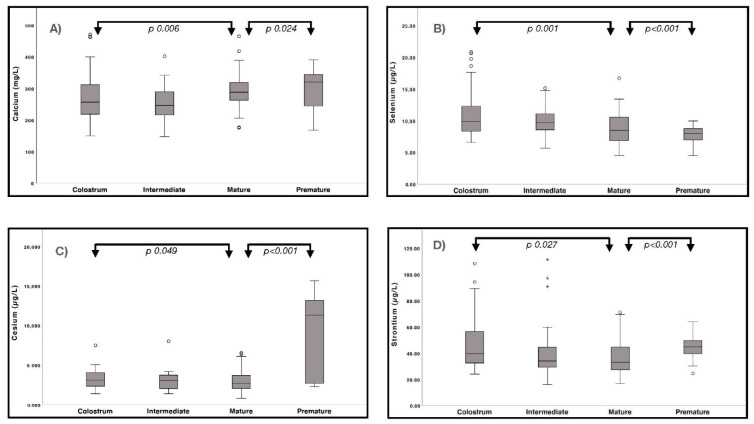
Box plots showing the distribution of calcium (**A**), selenium (**B**), cesium (**C**) and strontium (**D**) concentrations in human milk sampled from preterm mothers and from full-term mothers at three distinct stages of lactation. Box plots contains 50% of all values (25th to 75th percentile) with the median values indicated as a thick horizontal line. The whiskers represent the highest and lowest values, and the dots and asterisks the extreme values.

**Table 1 nutrients-13-01885-t001:** Characteristics of nursing mothers and infants.

	Full-Term Mothers (*n* = 70)		Pre-Term Mothers (*n* = 100)		
	Mean (± SD)/Number	Range/%	Mean ± SD/Number	Range/%	*p*
Mother’s age (y)	31.91 ± 4.58	24–44	35.52 ± 5.66	23–46	0.234
Maternal weight before pregnancy (kg)	69.74 ± 7.64	47–122	64.43 ± 6.69	48–92	0.424
Excessive weight gain during pregnancy (≥16 kg)	17	24%	23	23%	0.645
Multiple pregnancy	2	2%	13	13%	0.593
Gestational HT	5	7%	17	17%	0.283
Gestational diabetes	4	5%	10	10%	0.103
Residency (urban vs. rural)	52 vs. 18	75%	86 vs. 14	86%	0.248
Well water consumers	14	20%	27	27%	0.323
Smokers	10	14%	12	12%	0.548
Gestational age (wk)	39.12 ± 1.08	37–41.3	31.15 ± 3.25	24.33–34.66	0.043
Newborn weight (g)	2990 ± 391	2410–3830	1445 ± 554	670–2790	0.047

g, grams; HT, hypertension; kg, kilograms; SD, standard deviation; wk, weeks; y, years.

**Table 2 nutrients-13-01885-t002:** Concentration of trace elements in human milk according to lactation stage.

Element	Type of Milk	Mean ± SD	Interval for Mean		*p*-Value	Element	Mean ± SD	Interval for Mean		*p*-Value
			Lower	Upper				Lower	Upper	
**Minerals (mg/L)**										
Ca	TC	245.36 ± 49	233.43	257.29	***0.006***	Na	188.88 ±53	106.14	231.61	>0.999
	TI	270.59 ± 73	250.92	290.27			122.36 ± 87	99.05	145.68	
	TM	***291.04* ± *53***	278.09	303.99			124.93 ± 60	110.18	139.68	
	PM	***298.76* ± *57***	280.36	310.17	***0.024***		131.39 ± 34	124.48	138.30	>0.999
K	TC	333.46 ± 94	308.14	458.78	>0.999	p	134.85 ± 19	130.28	139.62	>0.999
	TI	350.06 ± 70	332.84	467.28			129.62 ± 25	122.85	136.39	
	TM	434.10 ± 119	405.31	462.89			128.39 ± 27	121.67	135.10	
	PM	373.65 ± 103	352.92	494.38	>0.999		125.31 ± 31	119.10	131.51	>0.999
Mg	TC	29.91 ± 7	28.19	41.62	>0.999					
	TI	35.98 ± 6	34.12	37.84						
	TM	38.19 ± 8	36.21	40.17						
	PM	37.36 ± 6	36.16	38.56	>0.999					
**Essential Trace Elements (µg/L)**										
Co	TC	0.057 ± 0.02	0.049	0.065	>0.999	Mn	***2.60* ± *3.50***	1.67	3.54	***0.039***
	TI	0.052 ± 0.06	0.037	0.067			1.74 ± 0.75	1.56	1.93	
	TM	0.044 ± 0.02	0.039	0.050			***1.68* ± *1.00***	1.44	1.92	
	PM	0.052 ± 0.01	0.049	0.056	0.827		1.99 ± 0.93	1.80	2.18	>0.999
Cr	TC	3.61 ± 0.99	3.38	3.85	>0.999	Mo	***1.88* ± *1.2***	1.47	2.29	***<0.001***
	TI	3.5 ±0.00	3.5	3.5			1.22 ± 1.98	0.82	1.76	
	TM	3.5 ± 0.00	3.5	3.5			***0.96* ± *1.16***	0.68	1.25	
	PM	***3.22* ± *1.04***	3.01	3.71	***0.013***		0.70 ± 1.17	0.47	0.94	>0.999
Cu	TC	339.34 ± 185	211.20	289.03	***0.029***	Se	***10.82* ± *3.41***	9.90	11.73	***0.001***
	TI	269.15 ± 135	236.02	302.29			9.91 ± 1.95	9.44	10.38	
	TM	250.11 ± 163	290.21	389.86			***8.87* ± *2.44***	8.28	9.47	
	PM	265.33 ± 71	251.03	279.63	>0.999		***4.97* ± *3.77***	4.22	9.72	***<0.001***
Fe	TC	187.70 ± 90	162.82	211.32	>0.999	Zn	1005.21 ± 1019	762.22	1248.20	>0.999
	TI	185.28 ± 78	166.09	204.46			1041.41 ± 911	797.25	1285.57	
	TM	176.51 ± 94	157.97	198.96			1237.76 ± 949	1004.30	1471.22	
	PM	***138.43* ± *83***	119.71	190.15	***0.016***		***558.95* ± *716***	316.06	1401.85	***<0.001***
I	TC	108.63 ± 51	94.96	122.3	>0.999					
	TI	121.95 ± 51	109.37	134.52						
	TM	127.98 ± 88	106.84	149.12						
	PM	95.18 ± 53	84.57	105.79	***0.045***					
**Toxic Trace Elements (µg/L)**										
Ag *	TC	0.10 ± 0.00	0.10	0.10	>0.999	Pb	0.51 ± 1.56	0.14	0.88	>0.999
	TI	0.10 ± 0.00	0.10	0.10			0.33 ± 0.38	0.22	0.43	
	TM	0.10 ± 0.00	0.10	0.10			0.30 ± 0.23	0.25	0.36	
	PM	0.10 ± 0.00	0.10	0.10	>0.999		***0.10* ± *0.01***	0.09	0.10	***0.004***
Al	TC	8.54 ± 3.12	7.79	9.28	0.736	Pt	***0.10* ± *0.24***	0.03	0.16	***0.025***
	TI	7.44 ± 4.05	6.35	8.52			0.05 ± 0.04	0.04	0.06	
	TM	7.29 ± 1.11	7.02	7.56			***0.05* ± *0.03***	0.04	0.06	
	PM	7.92 ± 4.38	7.04	8.79	>0.999		0.04 ± 0.01	0.04	0.04	>0.999
As	TC	0.93 ± 1.54	0.52	1.34	>0.999	Rb	***427.41* ± *130***	392.51	462.31	***0.013***
	TI	1.11 ± 0.171	0.70	1.51			448.36 ± 131	416.22	480.51	
	TM	1.37 ± 1.82	0.93	1.82			***519.64* ± *164***	480.4	558.94	
	PM	1.17 ± 0.60	1.05	1.29	>0.999		492.31 ± 97	472.95	511.66	>0.999
Ba	TC	4.02 ± 8.52	1.98	6.05	>0.999	Sb	0.07 ± 0.04	0.05	0.08	>0.999
	TI	3.77 ± 4.68	2.51	5.02			0.06 ± 0.02	0.06	0.07	
	TM	3.25 ± 2.45	2.65	3.85			0.06 ± 0.04	0.05	0.07	
	PM	***2.46* ± *1.07***	2.24	2.67	***0.047***		***0.10* ± *0.17***	0.06	0.13	***0.013***
Be *	TC	0.10 ± 0.00	0.10	0.10	>0.999	Sn	***0.09* ± *0.07***	0.07	0.11	***<0.001***
	TI	0.10 ± 0.00	0.10	0.10			0.07 ± 0.01	0.06	0.07	
	TM	0.10 ± 0.00	0.10	0.10			0.07 ± 0.00	0.07	0.07	
	PM	0.10 ± 0.00	0.10	0.10	>0.999		0.07 ± 0.00	0.07	0.07	>0.999
Cd	TC	0.18 ± 0.07	0.16	0.20	0.754	Sr	***45.42* ± *18***	40.34	50.50	***0.027***
	TI	0.16 ± 0.05	0.14	0.17			38.18 ± 16	34.33	42.03	
	TM	0.15 ± 0.20	0.10	0.20			36.36 ± 12	33.40	39.32	
	PM	***0.45* ± *0.49***	0.35	0.54	***<0.001***		***44.37* ± *7.95***	42.78	55.95	***<0.001***
Cs	TC	***5.48* ± *9.86***	2.84	8.12	***0.049***	Ti	36.78 ± 7.81	34.68	38.87	>0.999
	TI	4.17 ± 4.86	3.01	5.33			37.25 ± 13	34.02	40.48	
	TM	3.13 ± 1.73	2.71	3.55			40.90 ± 7.75	39.05	42.75	
	PM	***9.17* ± *5.00***	5.17	10.17	***<0.001***		***48.82* ± *23***	45.06	54.58	***<0.001***
Ga	TC	1.84 ± 0.37	1.74	1.94	>0.999	Tl	0.03 ± 0.01	0.03	0.03	>0.999
	TI	1.93 ± 0.61	1.77	2.08			0.03 ± 0.02	0.02	0.03	
	TM	2.08 ± 0.51	1.95	2.20			0.04 ± 0.05	0.02	0.05	
	PM	***2.21* ± *0.57***	2.10	2.33	***0.005***		0.04 ± 0.01	0.04	0.04	>0.999
Hg	TC	0.34 ± 0.18	0.29	0.39	>0.999	U *	0.004 ± 0.00	0.004	0.004	>0.999
	TI	0.32 ± 0.12	0.29	0.35			0.004 ± 0.00	0.004	0.004	
	TM	0.31 ± 0.08	0.29	0.33			0.004 ± 0.00	0.004	0.004	
	PM	***0.42* ± *0.31***	0.45	0.18	***0.019***		0.004 ± 0.00	0.004	0.004	>0.999
Li	TC	2.48 ± 4.47	1.28	3..8	>0.999	V *	0.05 ± 0.00	0.05	0.05	0.642
	TI	2.04 ± 2.99	1.32	2.75			0.05 ± 0.01	0.04	0.05	
	TM	1.66 ± 1.37	1.32	2.99			0.05 ± 0.00	0.05	0.05	
	PM	1.94 ± 1.69	1.61	2.28	>0.999		***0.05* ± *0.01***	0.05	0.06	***0.003***
Ni	TC	1.8 ± 0.00	1.80	1.80	>0.999					
	TI	2.18 ± 1.12	1.88	2.48						
	TM	2.35 ± 2.69	1.69	3.00						
	PM	1.89 ± 0.83	1.72	2.06	>0.999					

Elements in bold and italic indicate the type of formula statistically significant. * Values below the detection limit. Ag, silver; Al, aluminum; As, arsenic; Ba, barium; Be, beryllium; Ca, calcium; Cd, cadmium; Co, cobalt; Cr, chromium; Cs, cesium; Cu, copper; Fe, iron; Ga, gallium; Hg, mercury; I, iodine; K, potassium; Li, lithium; Mg, magnesium; Mn, manganese; Mo, molybdenum; Na, sodium; Ni, nickel; P, phosphorus; PM, preterm milk; Pb, lead; Pt, platinum; Rb, rubidium; Sb, antimony; Se, selenium; Sn, tin; Sr, strontium; TC, full-term colostrum; Ti, titanium; TI, full-term intermediate milk; TI, thallium; TM, full-term mature milk; U, uranium; V, vanadium; Zn, zinc.

**Table 3 nutrients-13-01885-t003:** Concentration of trace elements in infant formula according to type of formula.

Element	Type of Formula	Mean ± SD	Interval for Mean		*p*-Value	Element	Mean ± SD	Interval for Mean		*p*-Value
			Lower	Upper				Lower	Upper	
**Minerals (mg/L)**										
Ca	SF	419.93 ± 135	345.14	494.72	>0.999	Na	147.06 ± 27	131.72	162.40	>0.999
	CF	417.62 ± 30	391.96	443.28			161.12 ± 12	150.49	171.75	
	HF	430.2 ± 86	322.49	537.90			190.6 ± 62	113.57	267.72	
	PF	509.5 ± 64	393	626			218.5 ± 54	180	257	
K	SF	473.06 ± 43	448.92	497.21	>0.999	P	271.37 ± 12	253.67	281.59	>0.999
	CF	443.87 ± 9.26	436.12	451.62			284 ± 56	261.15	288.77	
	HF	522.8 ± 95	404.43	641.16			312 ± 61	235.32	367	
	PF	518 ± 49	432.28	662.71			319 ± 67	271	381.59	
Mg	SF	50.46 ± 7.21	46.46	54.46	>0.999					
	CF	53.12 ± 15	40.36	65.88						
	HF	59.8 ± 13	43.64	75.95						
	PF	67.5 ± 7.77	62	73						
**Essential trace elements (µg/L)**										
Co	SF	0.25 ± 0.13	0.17	0.33	***<0.001***	Mn	92.8 ± 70	53.58	133.11	***<0.001***
	CF	0.25 ± 0.10	0.16	0.34			57.35 ± 27	34.27	80.43	
	HF	***0.11* ± *0.06***	0.02	0.19			***172.37* ± *139***	37.9	341.22	
	PF	***0.71* ± *0.52***	0.34	1.08			60.19 ± 26	31.37	79.02	
Cr	SF	2.71 ± 0.99	2.16	3.26	***<0.001***	Mo	31.30 ± 17	21.80	40.80	***0.012***
	CF	2.35 ± 0.90	1.60	3.11			28.19 ± 4.89	24.10	32.28	
	HF	***5.40* ± *3.9***	0.54	10.27			***21.12* ± *12***	5.72	36.52	
	PF	4.5 ± 4.43	1.4	7.6			33.13 ± 0.58	27.86	38.40	
Cu	SF	383.03 ± 82	337.25	428.81	>0.999	Se	18.58 ± 4.74	15.95	21.21	>0.999
	CF	350.74 ± 63	297.81	403.68			17.46 ± 3.79	14.29	20.63	
	HF	397.33 ± 66	314.82	479.84			21.02 ± 3.96	16.10	25.94	
	PF	420.94 ± 23	206.96	634.91			19.74 ± 3.44	17.31	22.18	
Fe	SF	6069.33 ± 1264	5369.12	6769.54	***0.027***	Zn	4647.53 ± 888	4155.71	5139.35	>0.999
	CF	***8925.65* ± *503***	8505.13	9346.36			4910.37 ± 1070	4015.30	5805.44	
	HF	7280.2 ± 1855	4939.29	9621.10			5045.8 ± 1670	2970.98	7120.61	
	PF	6269 ± 394	2793.26	9814.03			6708 ± 748	6179	7237	
I	SF	133.03 ± 34	114.05	152.00	>0.999					
	CF	156.36 ± 18	141.07	171.05						
	HF	140.07 ± 6.52	131.96	148.17						
	PF	163.26 ± 19	149.24	177.28						
**Toxic trace elements (µg/L)**										
Ag *	SF	0.10 ± 0.00	0.10	0.10	0.543	Pb	0.37 ± 0.13	0.29	0.44	>0.999
	CF	0.10 ± 0.00	0.10	0.10			0.36 ± 0.22	0.17	0.54	
	HF	0.10 ± 0.00	0.10	0.10			0.33 ± 0.18	0.10	0.56	
	PF	0.10 ± 0.00	0.10	0.10			0.51 ± 0.28	0.3	0.7	
Al	SF	54.5 ± 27	39.20	69.79	>0.999	Pt *	0.12 ± 0.00	0.12	0.12	>0.999
	CF	47.07 ± 25	25.98	68.15			0.12 ± 0.00	0.12	0.12	
	HF	60.81 ± 45	4.77	116.85			0.12 ± 0.00	0.12	0.12	
	PF	37.44 ± 11	29.51	45.28			0.12 ± 0.00	0.12	0.12	
As	SF	0.49 ± 0.13	0.41	0.56	>0.999	Rb	287.13 ± 137	210.92	363.33	>0.999
	CF	1.39 ± 2	0.4	3.07			292.61 ± 105	204.02	381.19	
	HF	0.61 ± 0.42	0.08	1.14			164.09 ± 144	15.11	363.96	
	PF	0.48 ± 0.12	0.4	0.57			105.26 ± 65	59.56	151	
Ba *	SF	6.7 ± 0.00	6.7	6.7	>0.999	Sb	***0.79* ± *0.69***	0.41	1.18	***0.017***
	CF	6.7 ± 0.00	6.7	6.7			0.32 ± 0.27	0.09	0.55	
	HF	6.7 ± 0.00	6.7	6.7			0.53 ± 0.51	0.1	1.41	
	PF	6.7 ± 0.00	6.7	6.7			0.2 ± 0.14	0.1	0.3	
Be	SF	15.68 ± 5.51	12.62	18.73	>0.999	Sn	3.96 ± 2.96	0.76	5.16	***<0.001***
	CF	17.15 ± 4.5	13.33	20.98			3.14 ± 2.57	0.28	10.04	
	HF	12.75 ± 4.39	7.29	18.20			***19.52* ± *11***	0.76	46.2	
	PF	14.73 ± 4.75	12.68	16.79			0.92 ± 0.13	0.83	1.02	
Cd *	SF	0.06 ± 0.00	0.06	0.06	>0.999	Sr	145.86 ± 53	116.31	175.40	***<0.001***
	CF	0.06 ± 0.00	0.06	0.06			121.82 ± 28	97.73	145.90	
	HF	0.06 ± 0.00	0.06	0.06			133.61 ± 70	46.29	220.42	
	PF	0.06 ± 0.00	0.06	0.06			***355.01* ± *336***	117.01	593.02	
Cs	SF	0.88 ± 0.62	0.56	1.22	>0.999	Ti	49 ± 13	41.65	53.54	>0.999
	CF	0.78 ± 0.23	0.58	0.98			45.63 ± 8.44	38.57	52.70	
	HF	0.66 ± 0.45	0.09	1.23			47.16 ± 21	20.52	73.80	
	PF	0.32 ± 0.21	0.17	0.48			60.95 ± 21	44.46	77.44	
Ga	SF	2.49 ± 0.78	2.06	2.93	0.438	Tl	0.03 ± 0.01	0.02	0.04	>0.999
	CF	2.37 ± 0.55	1.90	2.83			0.03 ± 0.01	0.02	0.04	
	HF	2.23 ± 1.04	0.93	3.52			0.03 ± 0.02	0.006	0.06	
	PF	2.93 ± 0.98	2.24	3.63			0.08 ± 0.03	0.06	0.11	
Hg	SF	0.78 ± 0.45	0.49	1	>0.999	U	0.56 ± 0.32	0.37	0.004	***0.014***
	CF	0.75 ± 0.46	0.36	1.13			0.70 ± 0.58	0.21	0.19	
	HF	0.60 ± 0.21	0.33	0.89			0.64 ± 0.69	0.2	1.88	
	PF	0.66 ± 0.10	0.6	0.7			***0.94* ± *0.71***	0.44	1.45	
Li	SF	1.46 ± 0.63	1.11	1.81	>0.999	V	0.87 ± 0.28	0.71	1.03	***0.019***
	CF	1.52 ± 0.65	0.97	2.07			1.84 ± 0.48	1.43	2.25	
	HF	1.62 ± 0.89	052	2.73			4.62 ± 2.96	0.79	11.24	
	PF	1.61 ± 0.24	1.44	1.79			***6.15* ± *4.92***	0.57	9.28	
Ni	SF	5.71 ± 4.79	3.05	8.37	>0.999					
	CF	3.65 ± 1.27	2.58	4.71						
	HF	4.32 ± 1.53	2.41	6.23						
	PF	6.62 ± 4.5	3.4	9.8						

Elements in bold and italic indicate the type of formula statistically significant. * Values below the detection limit. Ag, silver; Al, aluminum; As, arsenic; Ba, barium; Be, beryllium; Ca, calcium; Cd, cadmium; CF, continuation formula; Co, cobalt; Cr, chromium; Cs, cesium; Cu, copper; Fe, iron; Ga, Gallium; HF, hydrolyzed formula; Hg, mercury; I, iodine; K, potassium; Li, lithium; Mg, magnesium; Mn, manganese; Mo, molybdenum; Na, sodium; Ni, nickel; P, phosphorus; Pb, lead; PF, preterm formula; Pt, platinum; Rb, rubidium; Sb, antimony; Se, selenium; SF, starter formula; Sn, tin; Sr, strontium; Ti, titanium; TI, thallium; U, uranium; V, vanadium; Zn, zinc.

**Table 4 nutrients-13-01885-t004:** Differences in concentration of trace elements in human milk versus infant milk formula.

Element	SF vs. TM		HF vs. TM		PF vs. PM		Element	SF vs. TM		HF vs. TM		PF vs. PM	
	% Difference	*p*-Value	% Difference	*p*-Value	% Difference	*p*-Value		% Difference	*p*-Value	% Difference	*p*-Value	% Difference	*p*-Value
**Minerals (mg/dL)**													
Ca	30	<0.001	32	<0.001	41	<0.001	Mg	24	<0.001	36	<0.001	44	<0.001
K	26	<0.001	32	0.004	27	0.063							
**Essential trace elements (µg/L)**													
Co	80	<0.001	45	0.062	90	<0.001	Mo	96	<0.001	95.4	<0.001	97	<0.001
Cr	28	0.073	35	<0.001	16	0.534	Se	52	<0.001	57	<0.001	94	0.008
Fe	96	<0.001	97	<0.001	99	<0.001	Zn	73	<0.001	75.4	<0.001	70.7	<0.001
Mn	98	<0.001	98	<0.001	96	0.008							
**Toxic trace elements (µg/L)**													
Al	86	<0.001	88	<0.001	78	<0.001	Sb	53	<0.001	52	<0.001	63	0.062
Be	43	<0.001	41	<0.001	42	<0.001	Sn	72	0.001	58	<0.001	8	>0.999
Hg	53	<0.001	46	0.634	36	0.084	Sr	75	<0.001	72	<0.001	87	<0.001
Ni	58	<0.001	45.6	0.068	71	0.006	U	98	<0.001	99	<0.001	99	<0.001
Rb	−56	0.001	−54	<0.001	−78	0.001	V	94	<0.001	62	<0.001	79	<0.001

Al, aluminum; Be, beryllium; Ca, calcium; Co, cobalt; Cr, chromium; Fe, iron; HF, hydrolyzed infant formula; Hg, mercury; K, potassium; Mg, magnesium; Mn, manganese; Mo, molybdenum; Ni, nickel; PF, preterm formula milk; PM, preterm milk; Rb, rubidium; Sb, antimony; Se, selenium; SF, starter infant formula; Sn, Tin; Sr, strontium; TM, full-term mature milk; U, uranium; V, vanadium; Zn, zinc.

**Table 5 nutrients-13-01885-t005:** Linear regression to estimate absolute change in mean levels of trace elements in human milk according to medical, social, environmental, and demographic variables.

Element	Excessive Maternal Weight Gain during Pregnancy		Baby’s Birth Body Weight		Gestational HT		Residence in Urban Area		Well Water Consumption		Smokers	
	Coefficient	*p*-Value	Coefficient	*p*-Value	Coefficient	*p*-Value	Coefficient	*p*-Value	Coefficient	*p*-Value	Coefficient	*p*-Value
**Minerals (mg/L)**												
Ca	−0.06	>0.999	−0.20	<0.001	5.257	0.024	0.257	>0.999	0.016	>0.999	0.429	>0.999
Mg	−0.12	>0.999	−0.19	0.001	0.776	>0.999	0.776	>0.999	2.524	>0.999	0.921	>0.999
Na	0.11	0.049	−0.23	<0.001	11.397	0.001	0.397	>0.999	6.133	0.015	0.314	>0.999
**Essential trace elements (µg/L)**												
Co	−0.01	>0.999	−0.17	0.002	0.015	>0.999	0.015	>0.999	0.57	>0.999	2.43	>0.999
Cr	−0.07	>0.999	−0.21	<0.001	0.127	>0.999	0.127	>0.999	1.72	>0.999	0	>0.999
Cu	−0.06	>0.999	−0.01	>0.999	0.475	>0.999	0.475	>0.999	4.24	0.041	0.05	>0.999
Fe	0.11	0.048	−0.08	>0.999	3.035	>0.999	3.035	>0.999	5.8	0.017	0.06	>0.999
I	0.11	0.042	−0.22	<0.001	1.906	0.063	1.906	>0.999	1.50	>0.999	0.78	>0.999
Mo	−0.07	0.024	−0.17	0.002	0.008	>0.999	0.008	>0.999	2.64	>0.999	0.22	>0.999
Se	−0.18	0.002	−0.28	<0.001	−0.11	0.049	1.11	>0.999	1.58	0.058	0.61	0.493
Zn	−0.17	>0.999	−0.30	<0.001	2.183	>0.999	2.183	>0.999	1.65	>0.999	0.21	>0.999
**Toxic trace elements (µg/L)**												
As	−0.05	>0.999	−0.24	<0.001	0.564	0.394	0.11	0.013	0.101	>0.294	0.558	>0.999
Ba	−0.01	>0.999	−0.04	>0.999	0.804	>0.999	0.804	>0.999	0	>0.999	7.96	0.049
Cs	−0.15	>0.999	−0.34	<0.001	0.502	>0.999	0.502	>0.999	0.71	>0.999	0.51	>0.999
Ga	−0.09	0.193	−0.14	0.016	3.75	0.047	0.75	>0.999	2.24	>0.999	0.025	>0.999
Hg	−0.12	0.032	−0.18	0.001	0.43	>0.999	0.43	>0.999	0.085	>0.999	2.42	0.094

Pb	−0.04	>0.999	0.45	<0.001	1.395	>0.999	1.395	0.075	3.674	0.046	9.61	0.024
Sb	0.002	>0.999	−0.16	0.006	0.178	>0.999	0.178	>0.999	1.37	0.057	0.453	>0.999
Sn	0.005	>0.999	0.17	0.004	1.224	0.082	1.224	>0.999	1.995	>0.999	0.188	>0.999
Sr	−0.08	0.027	−0.20	0.001	5.126	0.021	0.126	>0.999	2.595	>0.999	0.5	>0.999
Ti	−0.06	>0.999	−0.13	0.029	5.27	0.025	0.27	>0.999	2.997	0.041	0.52	>0.999
Tl	−0.005	0.938	−0.26	<0.001	0.017	>0.999	0.017	>0.999	2.601	>0.999	0.653	0.827
V	−0.03	>0.999	−0.27	<0.001	7.865	0.002	0.865	0.628	1.358	>0.999	0.699	>0.999

As, arsenic; Ba, barium; Ca, calcium; Co, cobalt; Cr, chromium; Cs, cesium; Cu, copper; Fe, iron; Ga, gallium; Hg, mercury; HT, hypertension; I, iodine; Mg, magnesium; Mo, molybdenum; Na, sodium; Pb, lead; Sb, antimony; Se, selenium; Sn, tin; Sr, strontium; Ti, titanium; TI, thallium; V, vanadium; Zn, zinc.

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
