# Peer review of "Human Milk Concentrations of Minerals, Essential and Toxic Trace Elements and Association with Selective Medical, Social, Demographic and Environmental Factors"

_nutrients, 2021, doi:10.3390/nu13061885_

Round 1

Reviewer 1 Report

A very good manuscript which is nicely written.

  1. Consider a small expansion of the Title which is currently more generic. A good example is "Human milk concentrations of minerals, essential and toxic trace elements and association with selective medical, social, demographic and environmental factors".
  2. Line #27 in the Abstract: Insert "were" before "associated with ...".
  3. Line # 72: Informed consent - define as "written or recorded oral informed consent or both"
  4. Figure 1: "Colostro" should be "Colostrum".
  5. For the 13 Starter infant formula, are they all milk protein based formulas or are there soy protein based or lactose-free milk protein based formulas? The extraction process for some of these alternative formulas could add significant levels of Al, Mn, etc.
  6. For the 5 hydrolyzed formulas, were they partially or extensively hydrolyzed formulas?
  7. Somewhere in the Study design and population, can you state the clinical trial registration agency or website and the # ?
  8. Statistical analysis section: Can you clarify whether the statistical method had an adjustment for unequal sample size when HM is compared with IF.
  9. Line 179: were the correlations positive or negative?
  10. Line 355: Consider adding as a study limitation, "the smaller sample size of the IF in this study.

Author Response

REVIEWER 1

  1. Consider a small expansion of the Title which is currently more generic. A good example is "Human milk concentrations of minerals, essential and toxic trace elements and association with selective medical, social, demographic and environmental factors”

ANSWER: We have changed the title according your suggestions. “Human milk concentrations of minerals, essential and toxic trace elements and association with selective medical, social, demographic and environmental factors”

  1. Line #27 in the Abstract: Insert "were" before "associated with ...".

ANSWER: The change was made

  1. Line # 72: Informed consent - define as "written or recorded oral informed consent or both"

ANSWER: All participating mothers signed the written informed consent.

  1. Figure 1: "Colostro" should be "Colostrum".

ANSWER: The change was made

  1. For the 13 Starter infant formula, are they all milk protein based formulas or are there soy protein based or lactose-free milk protein based formulas? The extraction process for some of these alternative formulas could add significant levels of Al, Mn, etc.

ANSWER: Both starter and continuation formulas are milk protein based formulas. We added this in the test

  1. For the 5 hydrolyzed formulas, were they partially or extensively hydrolyzed formulas?

ANSWER: We used one extensively hydrolyzed IF and 4 partially hydrolyzed IF, not observing significant differences between them.

  1. Somewhere in the Study design and population, can you state the clinical trial registration agency or website and the # ?

ANSWER: In the General Secretary of the Galician Health Council there is an active and updated registry of the studies approved in this región

  1. Statistical analysis section: Can you clarify whether the statistical method had an adjustment for unequal sample size when HM is compared with IF.

ANSWER: Thank you very much for your suggestion, we have forgot this in the statistical analysis. We added:

“In addition, in order to compare the means in groups of different sizes, the Bonferroni test was used”

.

  1. Line 179: were the correlations positive or negative?

ANSWER:. We observed significant positive correlations between HM concentrations of Ca and Na and gestational hypertension.

  1. Line 355: Consider adding as a study limitation, "the smaller sample size of the IF in this study.

ANSWER: Thank you very much for your suggestion. This data was added

Reviewer 2 Report

The authors attempt to characterise human milk in terms of its elemental content. A novelty of the work is the determination of the content of rare elements in milk and infant formula and comparison of the concentration of 35 elements (essential and toxic) between these type of products. There are a few details missing in the article (please see below), There is some information missing from the article, but having completed it, I recommend that the article be published in the Journal.

Abstract

The abstract is precise and states the main findings of the research presented in the article.

Introduction

The introduction do not provide sufficient background, therefore should be expanded. I propose adding a paragraph on the influence of the food consumed on the composition of human milk. I provide suggested literature below. A similar paragraph can be devoted to environmental, social and demographic factors.

  1. Multivariate analysis of essential elements in raw cocoa and processed chocolate mass materials from three different manufacturers, LWT, DOI  1016/j.lwt.2018.08.030
  2. Berry Fruits: Compositional Elements, Biochemical Activities, and the Impact of Their Intake on Human Health, Performance, and Disease, J. Agric. Food Chem., https://doi.org/10.1021/jf071988k
  3. Determination of Trace Elements in Meat and Fish Samples by MIP OES Using Solid-Phase Extraction, Food Analytical Methods, https://doi.org/10.1007/s12161-019-01615-3

Line 61: What did the authors mean by writing “…. in our health area, …”

lines 60-64: Was also the aim of the article to evaluate the composition of milk and infant formula in terms of health and potential health risks? Please consider whether you need to re-write the aim of your work.

Materials and methods

line 79: FI samples or you mean IF samples?

lines 106-107: perhaps: …plastic containers made of polyethylene …

The description of the test methodology for the determination of elements should be extended. line 116: Please provide the citation of literature describing the use ICP-OES for human milk and/or infant formula. If you don’t have any then please provide validation informations for this analytical method. It is also necessary to add informations on sample preparation for elemental analysis, such as the sample digestion procedure.

Please describe the procedure for comparing the elemental content between milk and infant formula. Which type of milk and which type of infant formula were taken for comparison presented in the table 4? Or perhaps comparisons were made on the average elemental content of these types of milk/infant formulas?

Results

The authors should describe the information obtained in Table 1. Please explain what the entries of the percentage ranges mean, e.g. 24-44, 47-122. Please refer to the statistics provided in the table 1.

Please provide better quality of box plots presented as figure 2. The font is too small to read the data in the graphs.

Please indicate in the tables the statistically different groups for content of each element in the milk or infant formula.

Discussion

lines 210-217: it is a repetition of information already given, I suggests deleting

This section is very well written, linking differences in elemental content with medical, environmental and other factors monitored in the mothers taking part in the study. Changes in elemental concentrations in relation to type of milk were considered. The content of elements in infant formula was compared with current recommendations. Also, the elemental results obtained were compared with the available literature.

Conclusions

Conclusions are drawn from the research results obtained and their discussion. They are relevant and demonstrate the novelty that the article brings to the state of current knowledge.

Author Response

REVIEWER 2

- Introduction:

1.The introduction do not provide sufficient background, therefore should be expanded. I propose adding a paragraph on the influence of the food consumed on the composition of human milk. I provide suggested literature below. A similar paragraph can be devoted to environmental, social and demographic factors.

ANSWER: According your suggestions, we add more background about food and environmental influence in breast milk:

The composition of HM is not always the same, the fat and energy content varies from the beginning to the end of the HM intake, it follows a diurnal pattern and varies between each individual, depending on the type of delivery, lactation period, maternal diet and area of residence (6, 7). It is widely reported that maternal diet influences the nutritional composition of breast milk (8). However, the amount of variability in HM attributable to diet remains mostly unknown. Previous studies on trace elements in HM included factors affecting its trace elements and maternal diet. Most studies have focused on component analysis or nutritional aspects of HM, but only a few studies have confirmed the relationship between trace elements in HM and psychosocial variables (9)”.

2.Line 61: What did the authors mean by writing “…. in our health area, …”

ANSWER: Our health area means the population served by our hospital. We have changed the term

3.Lines 60-64: Was also the aim of the article to evaluate the composition of milk and infant formula in terms of health and potential health risks? Please consider whether you need to re-write the aim of your work.

ANSWER: Thank you very much for you comment. We changed the objective according your suggestions:

“To evaluate the composition of milk in our population and potential health risks associated, we quantified levels of minerals and trace elements in HM samples acquired at different stages from mothers of preterm and full-term infants and in samples of IF for infants in the first year of life”.

Material and methods

4.Line 79: FI samples or you mean IF samples?

ANSWER: Sorry for the mistake, the change was made.

5.Lines 106-107: perhaps: …plastic containers made of polyethylene

ANSWER: This word was added.

  1. The description of the test methodology for the determination of elements should be extended. line 116: Please provide the citation of literature describing the use ICP-OES for human milk and/or infant formula. If you don’t have any then please provide validation informations for this analytical method. It is also necessary to add informations on sample preparation for elemental analysis, such as the sample digestion procedure.

ANSWER: You are right. We added:

Levels of trace elements in milk samples were quantified using inductively-coupled plasma mass spectrometry (ICP-MS), following the procedure described by Mohd-Taufek et al. (22). For this, a solution is prepared containing 0.01% (m/V) of Triton X-100, 10 g/L of Ethylenediaminetetraacetic Acid (EDTA), 2.5% (v/v) of ammonia and 10% (v/v) of 2-propanol prepared in Mili-Q® ultrapure water. Once the HM samples are homogenized by heating them in an ultrasonic bath between 35-38 °C, 400 µL of milk is taken, 1 mL of the previously prepared alkaline solution is added and it is brought to a final volume of 10 mL with Military H2O Q®.

The preparation of IF samples has been performed by simplifying the process, since the fat content of IF is lower than in HM and is also hydrolyzed, resulting in a much simpler matrix. The quantity of sample necessary to obtain the same proportion recommended by the manufacturer of the IF is weighed. Once dissolved and homogenized, the samples are taken 400 µL of milk and 1 mL of the solution of 0.01% (v/v) of Triton X-100 and H2O Mili-Q® is added to a final volume of 10 mL. The NIST SRM 1849 for IF samples and the certified milk reference materials ERM-BD 150 for low concentration levels and the ERM-BD 151 for higher levels in some elements have been used as certified reference materials. Once dissolved with ultrapure water, these materials have been prepared just like the samples.

The measurements of the trace elements in the milk samples have been performed with an ICP-MS model NexION® 300X (PerkinElmer Inc., Shelton, CT). The standard addition method has been used for the quantification of concentrations using different concentration levels between 0 to 25 µg/L. In the case of the majority elements, the measurement equipment has been used with an inductively coupled optical atomic emission spectrometry (ICP-OES) model Optima 3300DV (PerkinElmer Inc., Norwalk, USA). The calibration of the equipment has been carried out using the standard addition method with concentration standards between 0 and 5 mg/L for Ca, K, Mg and between 0-25 mg/L for Na and P. The instrumental conditions of the ICP-MS and ICP-OES are detailed in the Supplementary Table 1”.

  1. Please describe the procedure for comparing the elemental content between milk and infant formula. Which type of milk and which type of infant formula were taken for comparison presented in the table 4? Or perhaps comparisons were made on the average elemental content of these types of milk/infant formulas?

ANSWER: We compared mature human milk with starter and continuation infant formula, and preterm mature human milk with preterm formula. We explain it better in the text

- Results:

8.The authors should describe the information obtained in Table 1. Please explain what the entries of the percentage ranges mean, e.g. 24-44, 47-122. Please refer to the statistics provided in the table 1.

ANSWER: We explain the information of Table 1.

“Comparing the characteristics of the two groups, full-term mothers and preterm mothers, both are homogeneous in terms of no significant differences except in the mean gestational age (39 vs 31 weeks p <0.05) and the birth weight of the newborns (2990 g in term deliveries vs 1445 grams in preterm deliveries, p <0.05)”

9.Please provide better quality of box plots presented as figure 2. The font is too small to read the data in the graphs.

ANSWER: we try to improve.

10.Please indicate in the tables the statistically different groups for content of each element in the milk or infant formula.

ANSWER: We added this information. Elements in bold and italic indicate the type of formula statistically significant.

- Discussion:

11.Lines 210-217: it is a repetition of information already given, I suggests deleting.

ANSWER: We delete this according your suggestions.
